# HoxB-derived hoxba and hoxbb clusters are essential for the anterior–posterior positioning of zebrafish pectoral fins

Morimichi Kikuchi[†], Renka Fujii[†], Daiki Kobayashi[†], Yuki Kawabe, Haruna Kanno, Sohju Toyama, Farah Tawakkal, Kazuya Yamada, Akinori Kawamura*

Division of Life Science, Graduate School of Science and Engineering, Saitama University, Saitama, Japan

## eLife Assessment

This **important** study advances our understanding of vertebrate forelimb development, specifically the contribution of Hox genes to zebrafish pectoral fin formation. The authors have employed a robust and extensive genetic approach to tackle a key and unresolved question. The findings are overall **convincing** and will be of broad interest to developmental and evolutionary biologists.

*For correspondence:
akawamur@mail.saitama-u.ac.jp

[†]These authors contributed
equally to this work

Competing interest: The authors
declare that no competing
interests exist.

Reviewing Editor: Gerrit
Begemann, University of
Bayreuth, Germany

**Abstract** Vertebrate paired appendages, such as the pectoral fins in fish and the forelimbs in tetrapods, arise at specific regions along the anterior–posterior axis of the body. Hox genes have long been considered prime candidates for determining the anteroposterior positioning of these paired appendages during development. Evidence from various model organisms, including mouse and chick, supports a role for Hox genes in limb positioning. However, despite extensive phenotypic analyses of numerous single and compound Hox knockout mice, clear genetic evidence for substantial defects in limb positioning has been limited, leaving questions unresolved. In a previous study, we generated seven distinct hox cluster-deficient mutants in zebrafish. Here, we provide genetic evidence that zebrafish *hoxba;hoxbb* cluster-deleted mutants specifically exhibit a complete lack of pectoral fins, accompanied by the absence of *tbx5a* expression in pectoral fin buds. In these mutants, *tbx5a* expression in the pectoral fin field of the lateral plate mesoderm fails to be induced at an early stage, suggesting a loss of pectoral fin precursor cells. Furthermore, the competence to respond to retinoic acid is lost in *hoxba;hoxbb* cluster mutants, indicating that *tbx5a* expression cannot be induced in the pectoral fin buds. We further identify *hoxb4a*, *hoxb5a*, and *hoxb5b* as pivotal genes underlying this process. Although the frameshift mutations in these hox genes do not recapitulate the absence of pectoral fins, we demonstrate that deletion mutants at these genomic loci show the absence of pectoral fins with low penetrance. Our results suggest that, by establishing the expression domains along the anteroposterior axis, *hoxb4a*, *hoxb5a*, and *hoxb5b* within hoxba and hoxbb clusters cooperatively determine the positioning of zebrafish pectoral fins through the induction of *tbx5a* expression in the restricted pectoral fin field. Our findings also provide insights into the evolutionary origin of paired appendages in vertebrates.

## Introduction

In jawed vertebrates, paired appendages—such as the pectoral and pelvic fins in fish, and their homologous forelimbs and hindlimbs in tetrapods—develop at precise locations along the anterior–posterior axis of each species. These paired appendages arise from progenitor cells located in distinct regions of the lateral plate mesoderm (*Murata et al., 2010*; *Nishimoto and Logan, 2016*; *Shimada et al.,*

2013). The anteroposterior positioning of these paired appendages has long fascinated researchers, yet our understanding of the molecular mechanisms underlying this process remains limited.

In bilaterian animals, Hox genes—encoding evolutionarily conserved homeodomain-containing transcription factors—provide positional information and developmental timing along the anterior–posterior axis (*Iimura and Pourquié, 2007*; *Izpisúa-Belmonte and Duboule, 1992*; *Krumlauf, 1994*). A defining feature of Hox genes is their structural organization into Hox clusters, where multiple Hox genes are arranged in a precise order. Additionally, Hox clusters exhibit a distinctive phenomenon known as Hox collinearity, where the genomic arrangement of Hox genes correlates with specific developmental regions along the body axes (*Dollé et al., 1989*; *Duboule and Dollé, 1989*; *Graham et al., 1989*). In vertebrates, Hox clusters underwent divergence due to two rounds of whole-genome duplication early in vertebrate evolution (*Dehal and Boore, 2005*; *Ohno, 1970*), leading to the establishment of four distinct Hox clusters (HoxA, HoxB, HoxC, and HoxD), each consisting essentially of 1–13 paralogous groups in tetrapods. In contrast, teleost fishes experienced an additional teleost-specific whole-genome duplication, followed by the loss of a hox cluster, resulting in seven hox clusters in zebrafish (*Amores et al., 1998*; *Woltering and Durston, 2006*).

Multiple studies in chick and mouse have suggested that the initial anteroposterior position of limbs is regulated by Hox genes (*Tanaka, 2013*; *Tickle, 2015*). For example, the anterior boundaries of Hox gene expression domains were shown to align with future limb positions in chick embryos (*Burke et al., 1995*; *Cohn et al., 1997*). Experimental manipulations in avian embryos, including over-expression or interference of specific Hox genes, led to altered positions of forelimb buds (*Moreau et al., 2019*). In mice, *Hoxb5* knockout mutants exhibit a rostral shift of forelimb buds with incomplete penetrance (*Rancourt et al., 1995*), while other subtle shifts in limb positioning have also been reported in Hox mutants (*Royle et al., 2021*). Furthermore, loss of *Gdf11* in mice, which alters the expression of posterior Hox9–13 genes, causes posterior displacement of hindlimb buds, whereas ectopic *Gdf11* induces anterior shifts of hindlimb position (*Matsubara et al., 2017*). At the molecular level, Hox proteins directly bind to the Tbx5 limb enhancer and regulate its expression, providing a mechanistic link between Hox activity and forelimb initiation (*Minguillon et al., 2012*; *Nishimoto et al., 2014*). Together, these studies provide strong evidence across different model organisms that Hox genes contribute to the regulation of limb positioning. Nevertheless, despite the generation of numerous single and compound Hox knockout mice, no severe defects in the initial positioning of limb buds have been documented, and the precise mechanisms by which Hox genes specify limb position remain incompletely understood. This stands in contrast to the well-established role of Hox genes in limb patterning after limb bud formation, where paralogous group 9–13 genes in the HoxA and HoxD clusters cooperatively control the proximal–distal axis of developing limbs (*Boulet and Capecchi, 2004*; *Davis et al., 1995*; *Fromental-Ramain et al., 1996a*; *Fromental-Ramain et al., 1996b*; *Kmita et al., 2005*). Consequently, the exact role of Hox genes in defining the initial position of limb formation along the anterior–posterior axis remains unclear.

Using zebrafish, which belong to the vertebrate class Actinopterygii and are distinct from the tetrapods of the class Sarcopterygii, we previously generated mutants lacking each of the seven zebrafish hox clusters using the CRISPR–Cas9 method (*Yamada et al., 2021*). In our recent genetic analyses of hoxaa, hoxab, and hoxda clusters—orthologous to the mouse HoxA and HoxD clusters—we demonstrated that, similar to mice, these zebrafish hox clusters cooperatively play an essential role in the formation of the pectoral fins (*Ishizaka et al., 2024*), which are homologous to the forelimbs. In this study, we provide the first genetic evidence, to our knowledge, that Hox genes specify the positions of paired appendages in vertebrates. The double-deletion mutants of hoxba and hoxbb clusters, derived from the ancient HoxB cluster, exhibit a complete absence of pectoral fins due to the failure to express *tbx5a*. Despite incomplete penetrance, we propose a model in which *hoxb4a*, *hoxb5a*, and *hoxb5b* cooperatively provide positional cues along the anterior–posterior axis within the lateral plate mesoderm, thereby specifying the initial positions for fin bud formation through the induction of *tbx5a* in the pectoral fin field.

## Results

### Absence of pectoral fins in *hoxba;hoxbb* cluster-deleted zebrafish

In a previous study, we created seven individual hox cluster-deficient mutants in zebrafish using the CRISPR–Cas9 system (*Yamada et al., 2021*). Among these mutants, hoxba cluster-deleted embryos exhibited morphological abnormalities in their pectoral fins at 3 dpf (*Figure 1A–C*). To further investigate the pectoral fin phenotype in hoxba cluster mutants, we first analyzed the expression patterns of *Tbx5* orthologs. Zebrafish possess two paralogs, *tbx5a* and *tbx5b* (*Boyle-Anderson et al., 2022*). Among them, *tbx5a* plays a predominant role in the initial induction of pectoral fin buds in zebrafish (*Ahn et al., 2002*; *Garrity et al., 2002*; *Ng et al., 2002*). When comparing with *tbx5a* expression in wild-type embryos, we found that the *tbx5a* signal was reduced in the pectoral fin buds of hoxba cluster mutants (*Figure 1G, H*). Given that hoxba and hoxbb clusters originated from the ancestral HoxB cluster through teleost-specific whole-genome duplication (*Amores et al., 1998*), there may be functional redundancy between them. Surprisingly, we discovered that the simultaneous deletion of both hoxba and hoxbb clusters resulted in the complete absence of pectoral fins (*Figure 1A, F*). In contrast, pectoral fins were present in *hoxba⁻/⁻;hoxbb⁺/⁻* and *hoxba⁺/⁻;hoxbb⁻/⁻* mutants (*Figure 1D, E*), indicating that an allele from either hoxba or hoxbb cluster is sufficient for the pectoral fin formation. Moreover, all embryos lacking pectoral fins were identified as *hoxba;hoxbb* double homozygous mutants, with the expected penetrance ($n = 15/252$; 5.9%) being consistent with predictions based on Mendelian genetics (1/16 = 6.3 %). Furthermore, while *hoxba;hoxbb* cluster double homozygous mutants are embryonic lethal around 5 dpf, we could not detect any trace of pectoral fin development. Alongside this phenotype, the expression of *tbx5a* was significantly reduced to nearly undetectable levels in *hoxba;hoxbb* cluster mutants at 30 hpf (*Figure 1G–L*). The absence of pectoral fins in zebrafish *hoxba;hoxbb* cluster mutants sharply contrasts with the results of a previous study, which showed that mice lacking all HoxB genes except for *Hoxb13* of the HoxB cluster did not exhibit apparent abnormalities in their forelimbs (*Medina-Martínez et al., 2000*). Our genetic results suggest that the zebrafish hoxba and hoxbb clusters cooperatively play a crucial role in pectoral fin formation.

### Pectoral fin loss specific to *hoxba;hoxbb* cluster-deleted zebrafish

Multiple genetic studies in mice have demonstrated the functional redundancy of Hox genes across the four Hox clusters (*Horan et al., 1995*; *McIntyre et al., 2007*; *van den Akker et al., 2001*; *Wellik and Capecchi, 2003*; *Wellik et al., 2002*). To investigate whether the absence of pectoral fins is specific to *hoxba;hoxbb* cluster mutants, we created double-hox cluster mutants by crossing hoxba cluster mutants with six other hox cluster mutants in zebrafish. The combinatorial mutations of hoxba cluster with five other hox clusters, excluding hoxbb cluster, did not enhance the reduced expression of *tbx5a* (*Figure 2A–H*). Furthermore, we examined combined mutants of hox clusters, excluding hoxba and hoxbb. In *hoxca⁻/⁻;hoxcb⁻/⁻* cluster mutants, where the functions of HoxC genes are completely absent, the expression patterns of *tbx5a* in fin buds were indistinguishable from those of wild-type embryos (*Figure 2I*). Additionally, our previous study showed that *tbx5a* expression is unaffected in *hoxaa⁻/⁻;hoxab⁻/⁻;hoxda⁻/⁻* cluster mutants (*Ishizaka et al., 2024*), which lack all HoxA- and HoxD-related genes. The results of our deletion mutations of zebrafish hox clusters in various combinations emphasize that the diminished expression of *tbx5a* in the pectoral fin buds is evident only in hoxba;hoxbb cluster mutants. Taken together, in contrast to the functional redundancy of the four Hox clusters in mice, our genetic analysis indicates that the Hox clusters responsible for the specification of pectoral fin buds are restricted to the HoxB-derived hoxba and hoxbb clusters among the seven hox clusters in zebrafish.

### Loss of fin progenitors with posterior expansion of cardiac marker expression

To understand the absence of pectoral fin formation in *hoxba;hoxbb* cluster mutants, we examined the expression patterns of *tbx5a* during embryogenesis (*Figure 3A–F*). In wild-type embryos at the 10-somite stage, *tbx5a* expression appeared as bilateral symmetric stripes in the anterior lateral mesoderm, extending from the posterior midbrain to the second somite (*Figure 3A*; *Ahn et al., 2002*). These *tbx5a*-positive cells subsequently divided into two groups: the anteriorly migrating cells form the heart primordia, while the posteriorly migrating cells give rise to the future pectoral

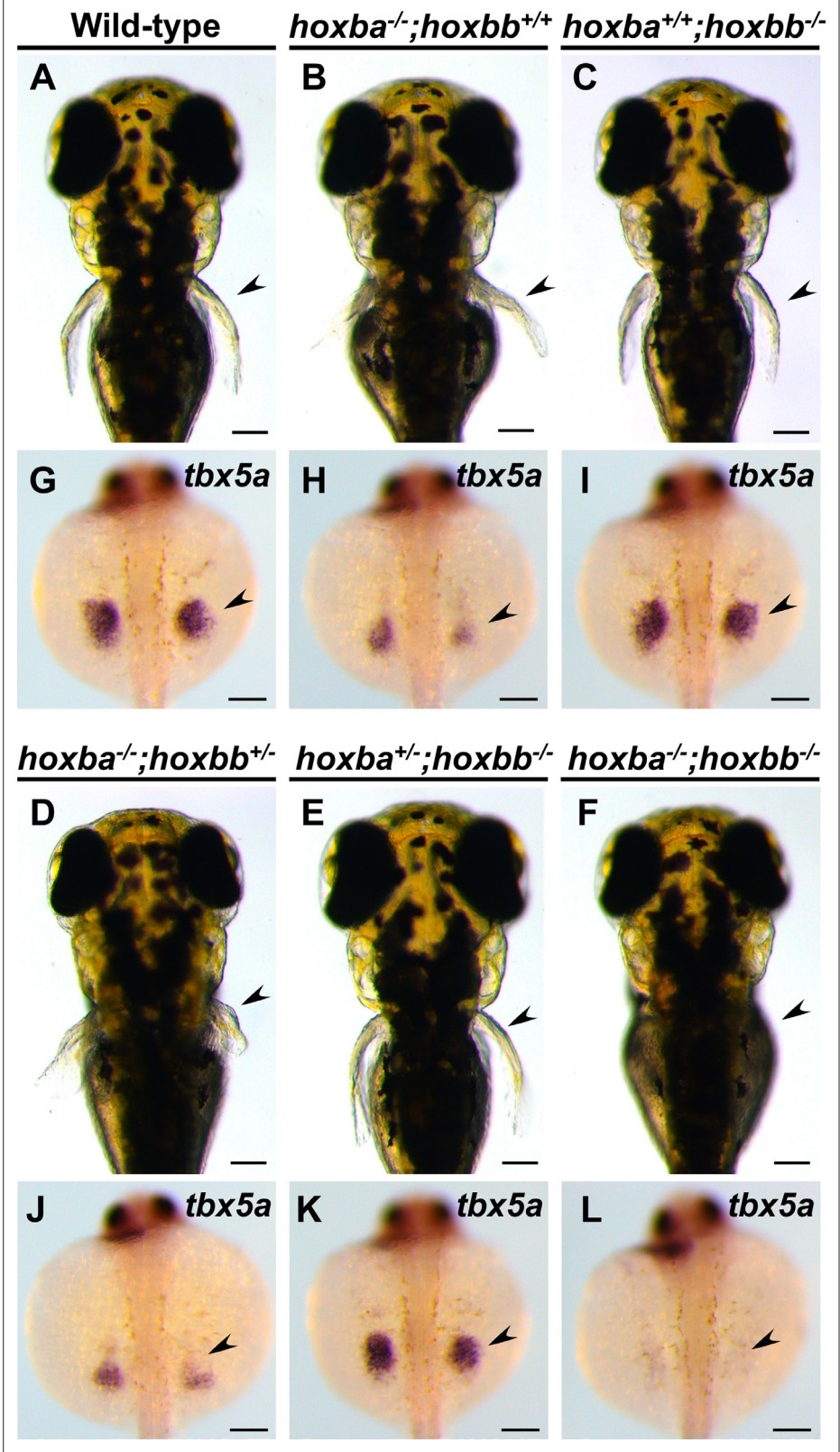

**Figure 1.** Lack of pectoral fins in *hoxba;hoxbb* cluster-deleted mutants. (**A–G**) Dorsal views of live zebrafish larvae at 3 dpf, obtained from intercrosses between *hoxba;hoxbb* cluster-deficient hemizygous fish. Arrowheads indicate the positions of the pectoral fins. (**H–L**) Expression patterns of *tbx5a* in the pectoral fin bud (arrowhead) at 30 hpf. Dorsal views are shown, and the genotype of each specimen was determined. For each genotype, reproducibility

*Figure 1 continued on next page*

*Figure 1 continued*

was confirmed with at least three different specimens. Genotyping revealed that all the embryos lacking pectoral fins (*n* = 15) were *hoxba;hoxbb* double homozygotes. Scale bars are 100 μm.

fin buds (*Figure 3C, E*). In contrast, abnormal expression patterns of *tbx5a* were evident at early stages in *hoxba;hoxbb* cluster mutants. Compared to wild-type embryos, the bilateral stripes of *tbx5a* expression were shortened along the anterior–posterior axis, and the *tbx5a*-positive signal was absent from the posterior region where pectoral fin progenitor cells typically emerge (*Figure 3B*). Although

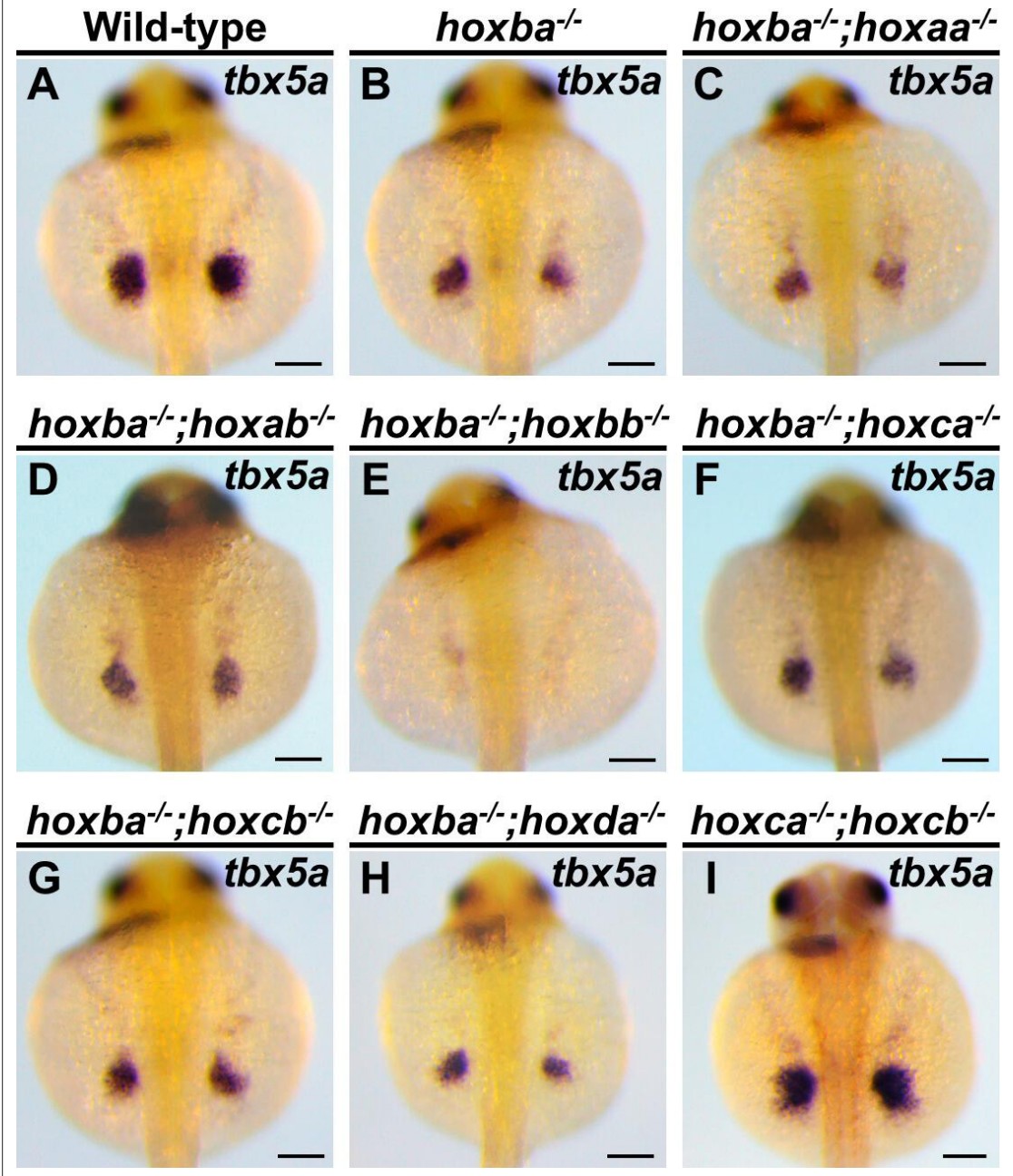

**Figure 2.** Significantly decreased expression of *tbx5a* in the pectoral fin buds is specific to *hoxba;hoxbb* cluster-deleted mutants. (A–I) Expression patterns of *tbx5a* in the pectoral fin buds of combinatorial deletion mutants of zebrafish *hox* clusters. Dorsal views of embryos at 30 hpf are displayed. After capturing images, the genotype of each specimen was determined. For each genotype, reproducibility was confirmed with at least three different specimens. Scale bars are 100 μm.

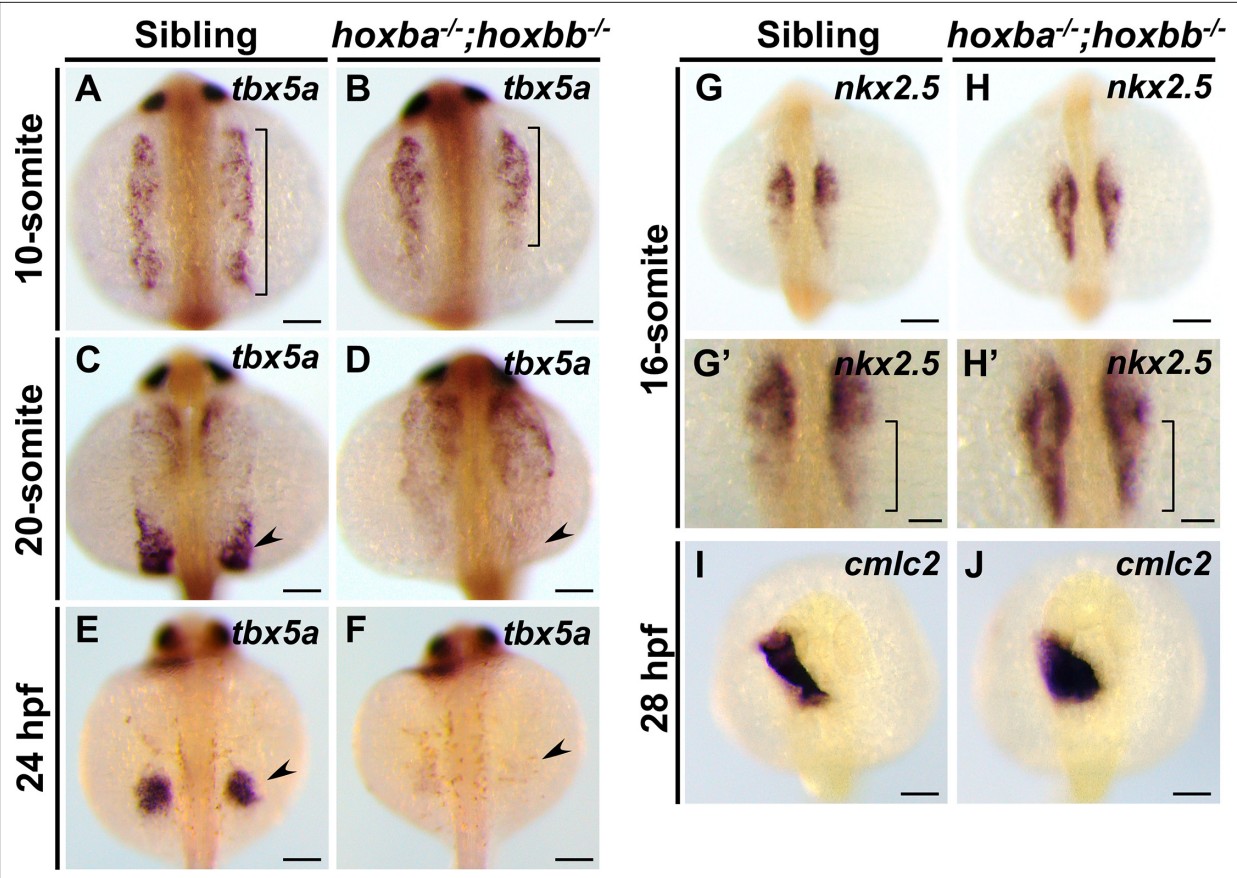

**Figure 3.** Loss of fin progenitors with posterior expansion of cardiac marker expression in *hoxba;hoxbb* mutants. (**A–F**) Expression patterns of *tbx5a* were compared between sibling wild-type and *hoxba;hoxbb* homozygous embryos during embryogenesis. The range of *tbx5a* expression in the lateral mesoderm is indicated by a bracket. The progenitor cells of pectoral fins are indicated by an arrowhead. (**G, H**) Expression patterns of *nkx2.5* were compared between sibling and *hoxba;hoxbb* mutants. Enlarged views are shown in (**G', H'**). The different regions of *nkx2.5* expression between wild-type and mutants are indicated by brackets. (**I, J**) Expression patterns of *cmlc2* are shown. All images were captured from the dorsal side. For each stage, reproducibility was confirmed with at least three different specimens. Scale bars indicate 100 µm.

anterior migration toward the presumptive heart occurred, posterior migration toward the fin buds was undetectable, consistent with the loss of pectoral fin progenitor cells in *hoxba;hoxbb* cluster mutants (*Figure 3D, F*). These results underscore the early role of the hoxba and hoxbb clusters in the formation of pectoral fin buds through the induction of *tbx5a* expression. In zebrafish, studies have shown a reciprocal relationship between pectoral fin and heart progenitor populations in the anterior lateral mesoderm, where retinoic acid (RA) signaling promotes pectoral fin formation while restricting cardiac progenitors (*Waxman et al., 2008*). Therefore, we also analyzed cardiac development (*Figure 3G–J*). By examining the expression patterns of *nkx2.5*, an early cardiac progenitor marker (*Chen and Fishman, 1996*; *Serbedzija et al., 1998*), we found that the expression domain of *nkx2.5* in *hoxba;hoxbb* cluster mutants extended more posteriorly compared to wild-type embryos (*Figure 3G, H*). Furthermore, the region positive for the differentiated cardiac cell-specific myosin light chain (*cmlc2*) showed abnormal morphology relative to wild-type embryos (*Figure 3I, J*; *Yelon et al., 1999*). Taken together, these results indicate that hoxba and hoxbb clusters are essential for establishing the pectoral fin field in zebrafish. In their absence, pectoral fin progenitor cells fail to emerge, accompanied by a posterior expansion of cardiac marker expression domain. Our results highlight a shift in the balance between fin and cardiac progenitor populations within the anterior lateral plate mesoderm.

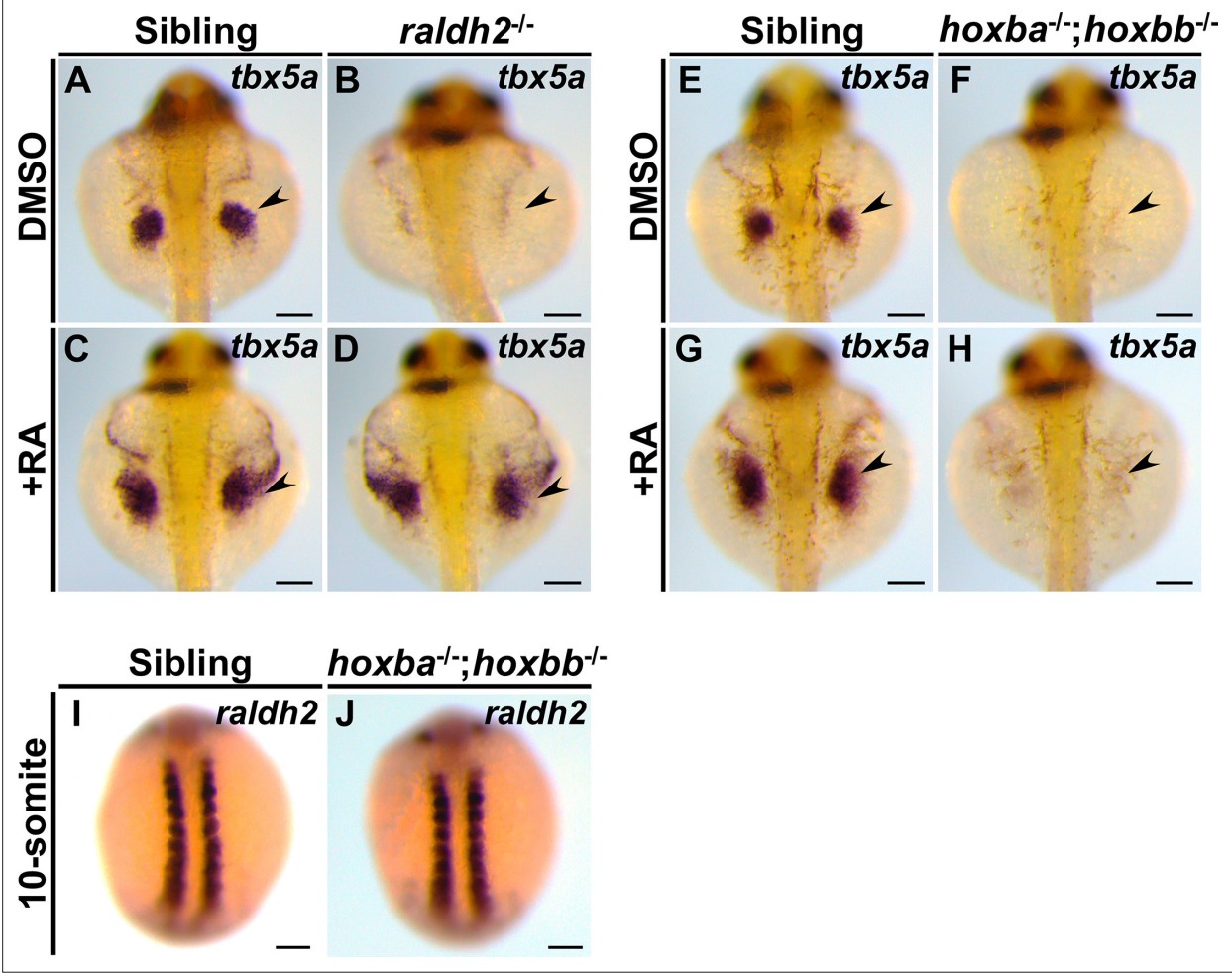

**Figure 4.** Zebrafish *hoxba;hoxbb* mutants lack a response to retinoic acid (RA) in fin buds. (**A–D**) Exogenous RA exposure in *raldh2* mutants can rescue *tbx5a* expression in fin buds. Arrowheads indicate the positions of the pectoral fin buds. (**E–H**) RA treatments in *hoxba;hoxbb* cluster-deleted mutants do not rescue *tbx5a* expression in fin buds. (**I, J**) Expression patterns of *raldh2* were compared between sibling and *hoxba;hoxbb* mutants. For each genotype, reproducibility was confirmed with at least three different specimens. All images were captured from the dorsal side. Scale bars represent 100 µm.

The online version of this article includes the following figure supplement(s) for figure 4:

**Figure supplement 1.** Generation of the zebrafish *raldh2* mutants by CRISPR–Cas9.

## hoxba/bb clusters required for RA-mediated *tbx5a* induction in the fin buds

RA is a well-known upstream regulator of Hox genes in vertebrates (*Boncinelli et al., 1991*; *Langston and Gudas, 1994*; *Marshall et al., 1996*). Loss of function of *retinaldehyde dehydrogenase 2* (*raldh2*), an essential regulator in RA synthesis, leads to the absence of pectoral fins in zebrafish (*Begemann et al., 2001*; *Grandel et al., 2002*), a phenotype similar to that observed in *hoxba;hoxbb* cluster mutants. Previous studies have shown that exogenous RA treatments can rescue pectoral fin formation in *raldh2⁻ᐟ⁻* embryos (*Gibert et al., 2006*; *Grandel et al., 2002*). It has been suggested that RA directly or indirectly induces *tbx5a* expression in zebrafish fin buds (*Gibert et al., 2006*; *Grandel and Brand, 2011*; *Neto et al., 2012*). The phenotypic similarities between *hoxba;hoxbb* cluster mutants and *raldh2* mutants prompted us to investigate whether RA exposure could also rescue the absence of pectoral fins in *hoxba;hoxbb* cluster mutants. To explore this, we introduced a frameshift mutation in *raldh2* using CRISPR–Cas9, confirming the phenotypic recapitulation of previously described *raldh2⁻ᐟ⁻* mutants (*Figure 4—figure supplement 1*). Consistent with prior results (*Gibert et al., 2006*; *Grandel et al., 2002*), RA treatment in our *raldh2* mutants rescued the expression of *tbx5a* in pectoral

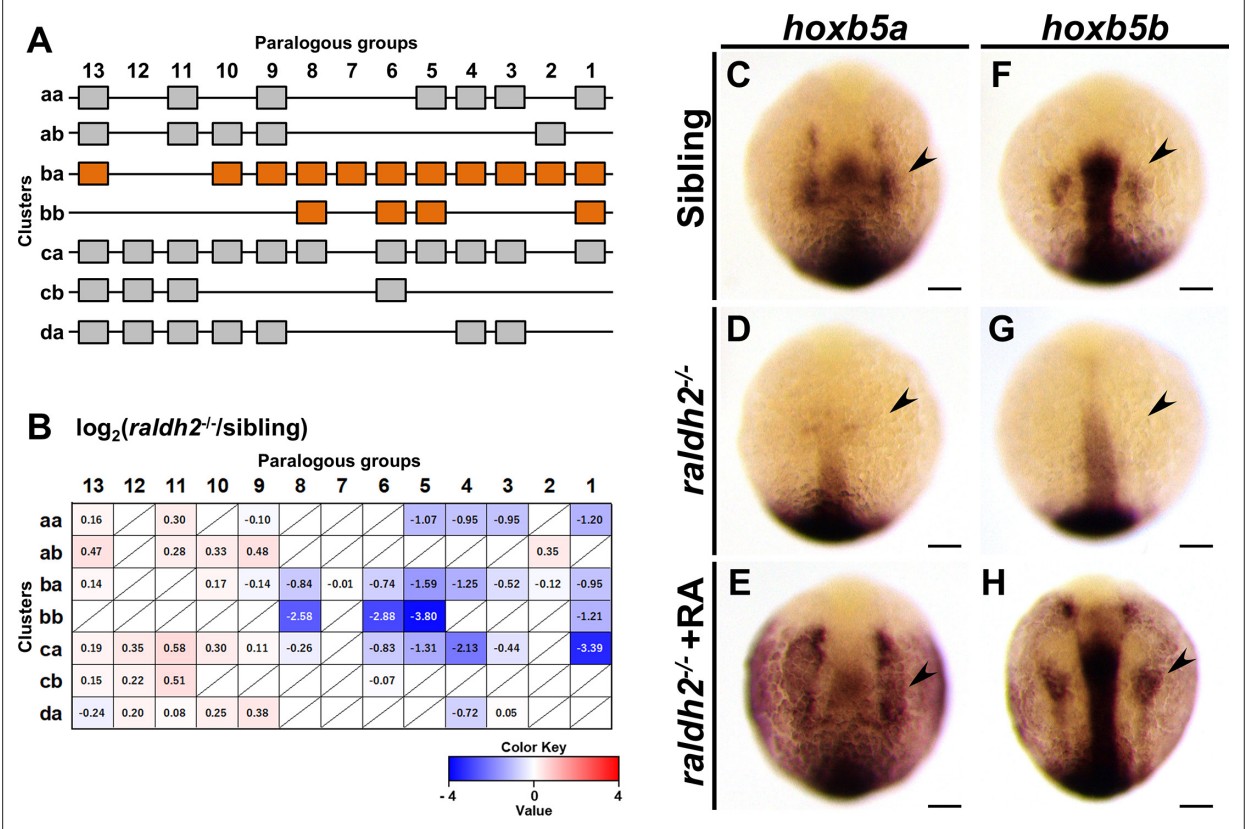

**Figure 5.** Expression patterns of *hox5a* and *hoxb5b* are regulated by RA. (**A**) Schematic representation of the 49 *hox* genes organized into seven *hox* clusters in zebrafish. *hox* genes in *hoxba* and *hoxbb* clusters are highlighted with orange. (**B**) Expression profiles of the 49 *hox* genes in wild-type and *raldh2⁻/⁻* embryos at the 20-somite stage, analyzed by RNA-seq. The average FPKM of each *hox* gene was compared between sibling and *raldh2* mutants. The absence of specific *hox* genes is indicated by a slash. (**C–H**) Expression patterns of *hoxb5a* and *hoxb5b* in sibling wild-type, *raldh2⁻/⁻*, and RA-treated *raldh2⁻/⁻* embryos at the 10-somite stage. Arrowheads indicate the presumptive positions of pectoral fin buds. For each genotype, reproducibility was confirmed with at least three different specimens. All images are captured from the dorsal side. Scale bars represent 100 μm.

The online version of this article includes the following source data and figure supplement(s) for figure 5:

**Source data 1.** This source data file contains the numerical values used to generate *Figure 5B*.

**Figure supplement 1.** Volcano plot of the transcriptome analysis between sibling and *raldh2* mutants.

**Figure supplement 1—source data 1.** This source data file provides the underlying numerical data used to create the volcano plot.

fins (*Figure 4A–D*). In contrast, RA did not induce *tbx5a* expression in the fin buds of *hoxba;hoxbb* cluster mutants (*Figure 4E–H*), and no trace of pectoral fin formation was observed in the treated embryos even at 5 dpf. Furthermore, the endogenous expression patterns of *raldh2* in *hoxba;hoxbb* cluster mutants seem indistinguishable from those in wild-type embryos (*Figure 4I, J*), suggesting that the response to RA is absent in the fin buds of *hoxba;hoxbb* cluster mutants, while RA synthesis appears intact. Although a model proposing that RA directly induces *tbx5a* expression has been suggested (*Gibert et al., 2006*; *Grandel and Brand, 2011*; *Neto et al., 2012*), our results indicate that RA-dependent induction of *tbx5a* does not occur in the absence of hoxba and hoxbb clusters. Given that *hoxba;hoxbb* cluster mutants retain the other five intact hox clusters, our findings further support the notion that the hoxba and hoxbb clusters are specifically responsible for the RA-mediated specification of pectoral fins.

### *hoxb5a/b5b* expression in fin buds is RA dependent

The absence of pectoral fins, commonly observed in *hoxba;hoxbb* cluster mutants and *raldh2* mutants, suggests that candidate hox genes may be downregulated in *raldh2* mutants. To investigate this, we performed RNA-seq analysis to compare the expression profiles between sibling and *raldh2* homozygous embryos. As expected, we found that most of the 3′-located hox genes are downregulated

in *raldh2*$^{-/-}$ embryos (*Figure 5A, B*, *Figure 5—figure supplement 1*). Although *hoxc1a* showed the most pronounced reduction, our analysis focused on *hoxba* and *hoxbb* clusters. Notably, the levels of *hoxb5b* transcripts were significantly reduced, along with those of *hoxb5a* among hoxba genes, in *raldh2* mutants. Previous studies have shown that zebrafish *hoxb5a* and *hoxb5b* are expressed in the lateral plate mesoderm corresponding to presumptive pectoral fin buds (*Waxman et al., 2008*). Additionally, knockout mice for *Hoxb5*, which is homologous to zebrafish *hoxb5a* and *hoxb5b*, only exhibit altered positioning of forelimbs (*Rancourt et al., 1995*), among various other Hox knockouts that have been generated. Therefore, we examined whether RA regulates the expression of *hoxb5a* and *hoxb5b* in fin buds. In wild-type embryos, *hoxb5a* expression was observed in the neural tube, somites, and lateral plate mesoderm (*Figure 5C*). However, in *raldh2* mutants, *hoxb5a* expression was significantly reduced, notably in the lateral plate mesoderm, which differentiates into the fin buds, where it was reduced to the point of being undetectable (*Figure 5D*). Since RA treatments in *raldh2* mutants can rescue pectoral fin development, we found that *hoxb5a* expression in *raldh2* mutants is also rescued by RA exposure (*Figure 5C–E*). Regarding the expression of *hoxb5b*, which diverged from the same ancestral gene as *hoxb5a*, RA-dependent expression was also confirmed, mirroring the observations made with *hoxb5a* (*Figure 5F–H*). These results suggest that both *hoxb5a* and *hoxb5b* are involved in the formation of the pectoral fins in zebrafish.

## Genetic screen for hox genes specifying pectoral fin buds

To understand the molecular mechanisms underlying pectoral fin specification, we sought to identify hox genes essential for fin formation through a genetic approach. We anticipated that frameshift mutations in relevant hox genes would phenocopy the absence of pectoral fins observed in *hoxba;hoxbb* mutants. Since the hoxbb cluster contains fewer hox genes than the hoxba cluster (*Figure 5A*), we focused our investigation on the hoxbb cluster. Although *hoxb5b* is expressed in the presumptive fin buds (*Figure 5F*), other hoxbb genes may also be involved. Therefore, we simultaneously injected three gRNAs targeting *hoxb5b*, *hoxb6b*, and *hoxb8b* into hoxba hemizygous mutants, expecting various germline mutations in these hoxbb genes in the injected fish (*Figure 6A*). Subsequently, F1 fish with a hemizygous deletion of the hoxba cluster and mosaic mutations were crossed with *hoxba;hoxbb* hemizygous mutants. Among the progeny, several embryos lacking pectoral fins were identified. Following genotyping and DNA sequencing of the target hoxbb genes, we found that all embryos without pectoral fins possessed frameshift mutations in *hoxb5b*, with *hoxba*$^{-/-}$;*hoxbb*$^{+/-}$ being common among them (*Figure 6B–E*, *Supplementary file 1A*, *Figure 6—figure supplement 1*). Similarly, we injected each crRNA for *hoxb6b* and *hoxb8b* into *hoxba*$^{+/-}$ mutants and confirmed that neither frameshift mutation in *hoxb6b* nor *hoxb8b* in *hoxba*$^{-/-}$;*hoxbb*$^{+/-}$ led to the loss of pectoral fins (*Figure 6F, G*, *Figure 6—figure supplement 1*). These results suggest that the frameshift mutation of *hoxb5b* alone can phenocopy the deletion of hoxbb cluster, recapitulating the absence of pectoral fins in *hoxba;hoxbb* cluster mutants.

However, we encountered an unexpected result: *hoxba*$^{-/-}$;*hoxb5b*$^{-/-}$ embryos, which have a homozygous frameshift mutation in *hoxb5b* and a homozygous deletion of hoxba cluster, exhibited severely truncated but visibly present pectoral fins (*Figure 6H, I*). After analyzing hundreds of embryos from several crossings, we did not find any *hoxba*$^{-/-}$;*hoxb5b*$^{-/-}$ embryos without pectoral fins. Consistent with this, the expression of *tbx5a* was detectable in the fin buds of *hoxba*$^{-/-}$;*hoxb5b*$^{-/-}$ embryos (*Figure 6K, L*), in contrast to its absence in *hoxba;hoxbb* cluster mutants (*Figure 1L*). To examine the potential contributions of other hoxbb genes, we observed the phenotypes of *hoxba*$^{-/-}$;*hoxb5b*$^{-/-}$; *hoxb6b*$^{-/-}$ embryos; however, these mutants did not exacerbate the abnormalities and still formed pectoral fins (*Figure 6J and M*).

The phenotype of *hoxba*$^{-/-}$;*hoxb5b*$^{-/-}$ embryos indicates that a frameshift mutation in *hoxb5b* does not fully mimic the complete deletion of the entire hoxbb cluster, suggesting that other mechanisms may be involved. A novel genetic compensation mechanism known as transcriptional adaptation has been identified: mRNA containing a premature termination codon (PTC) can induce increased expression of structurally related genes, thereby compensating for the function of mutated genes (*El-Brolosy et al., 2019*; *Rossi et al., 2015*). To avoid producing transcripts with PTC (*Sztal and Stainier, 2020*), we generated a full locus-deleted allele of *hoxb5b* using CRISPR–Cas9 with two crRNAs targeting both ends of the target locus (*Figure 7—figure supplement 1*). In contrast to the phenotype of *hoxba*$^{-/-}$;*hoxb5b*$^{-/-}$, we found that *hoxba*$^{-/-}$;*hoxb5b*$^{de/dell}$ mutants lack pectoral fins and exhibit

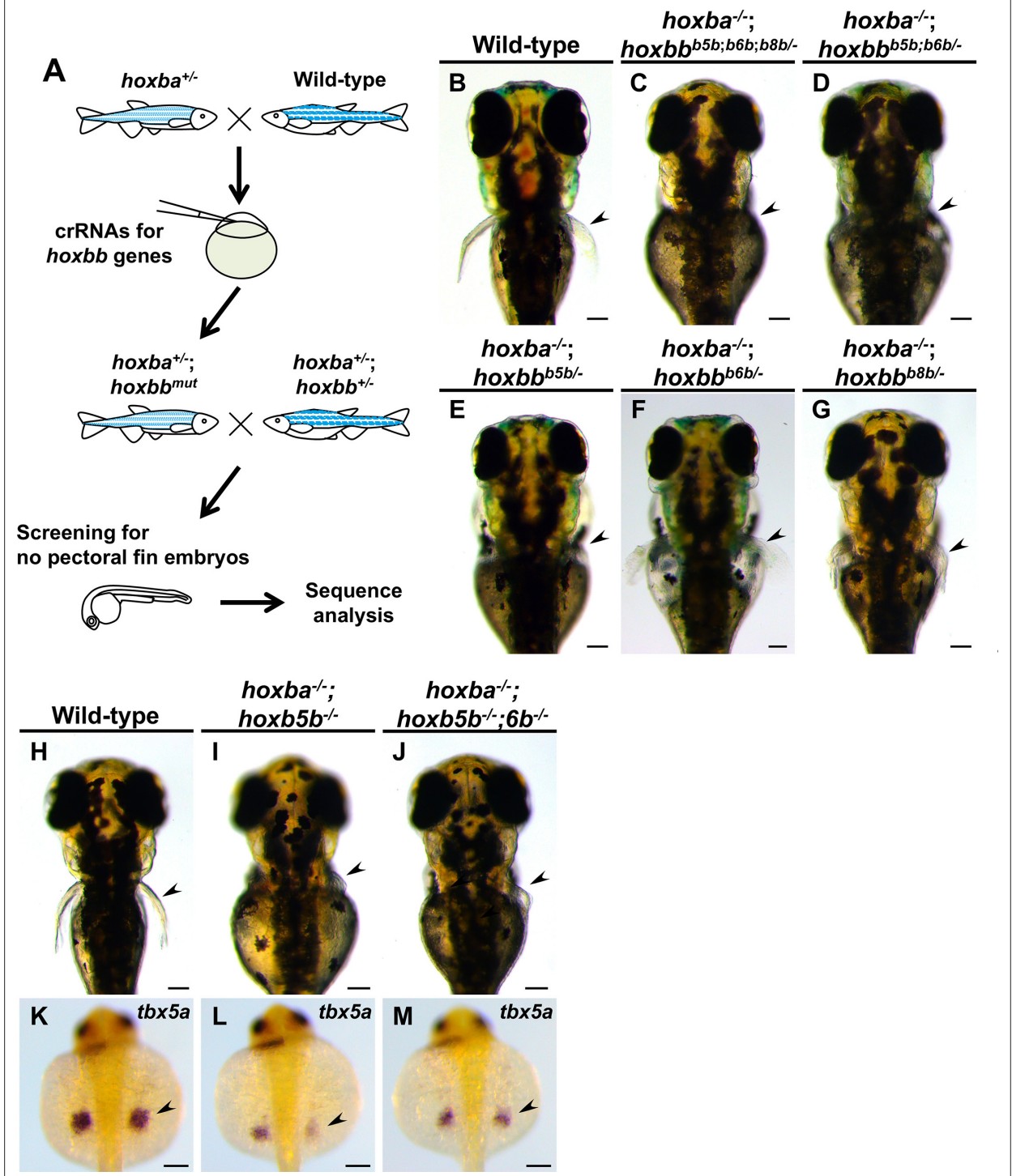

**Figure 6.** Screening for hox genes responsible for the zebrafish pectoral fin formation. (**A**) Schematic representation of the genetic screening for hox genes involved in the specification of pectoral fins in zebrafish. (**B–G**) Dorsal views of live mutant larvae at 3 dpf were obtained during the screening. After capturing images, genotyping, and DNA sequencing in target hoxb genes were conducted. For the mutants illustrated in (**C–G**), hoxb genes are shown with frameshift mutations introduced. Detailed information is provided in ***Supplementary file 1A***. (**H–J**) Dorsal views of hoxba$^{-/-}$;hoxb5b$^{-/-}$ and hoxba$^{-/-}$;hoxb5b$^{-/-}$;hoxb6b$^{-/-}$ larvae at 3 dpf. (**K–M**) Expression patterns of tbx5a in hoxba$^{-/-}$;hoxb5b$^{-/-}$ and hoxba$^{-/-}$;hoxb5b$^{-/-}$;hoxb6b$^{-/-}$ mutants at 30 hpf. Arrowheads indicate the presumptive positions of pectoral fin buds. All images are captured from the dorsal side. Scale bars represent 100 μm.

The online version of this article includes the following figure supplement(s) for figure 6:

**Figure supplement 1.** Generation of the frameshift-induced hox mutants using CRISPR–Cas9.

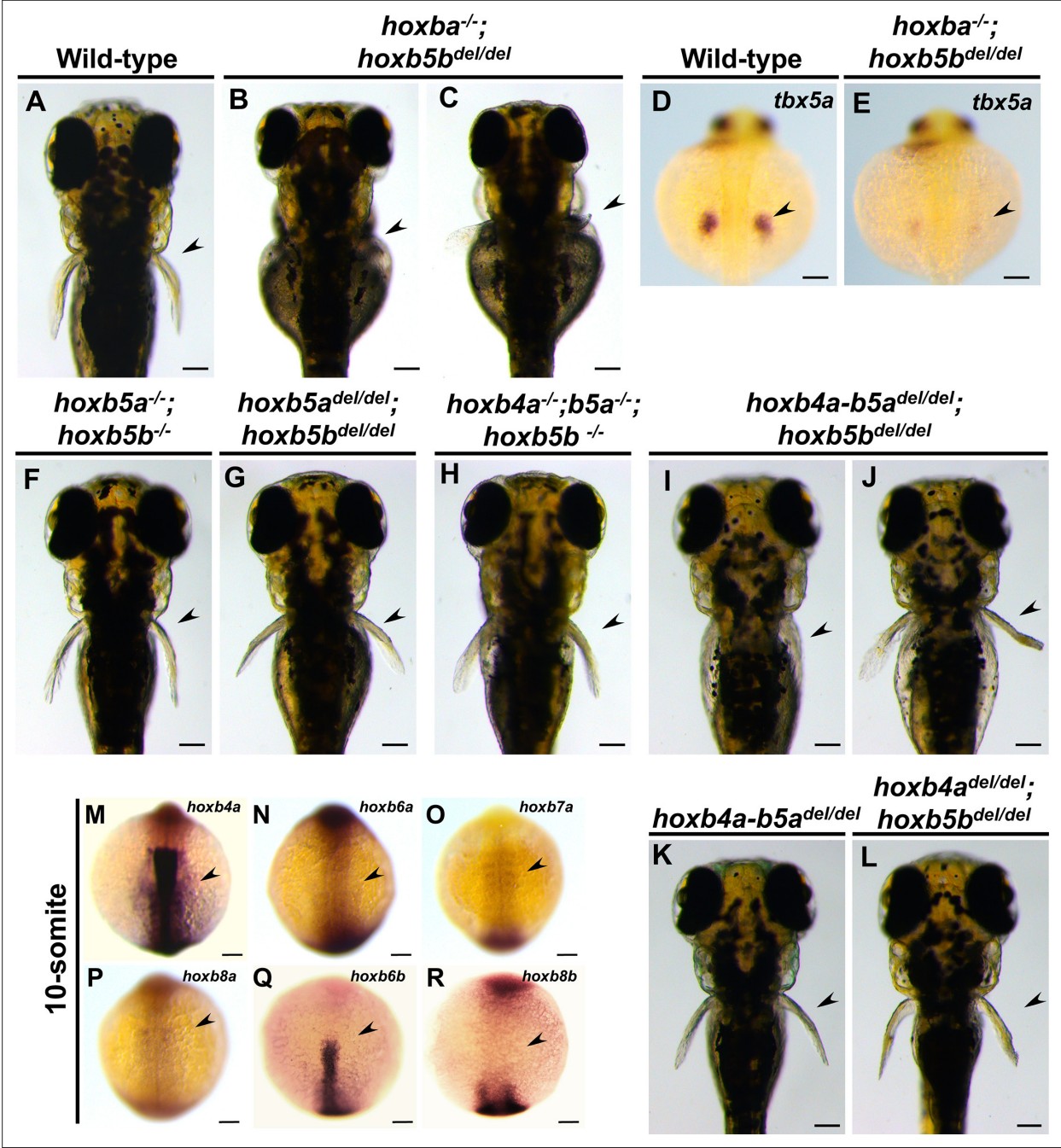

**Figure 7.** *hoxb4a-b5a^{del/del};hoxb5b^{del/del}* larvae partially recapitulate the absence of the pectoral fins. (**A–C**) Dorsal views of zebrafish *hoxba^{−/−};hoxb5b^{del/del}* larvae at 3 dpf. (**D, E**) Expression patterns of *tbx5a* in sibling wild-type and *hoxba^{−/−};hoxb5b^{del/del}* (*n* = 4) at 30 hpf. Arrowheads indicate the presumptive positions of pectoral fin buds. (**F, G**) Dorsal view of frameshift-induced *hoxb5a^{−/−};hoxb5b^{−/−}* and *hoxb5a^{del/del};hoxb5b^{del/del}* larvae. (**H–L**) Dorsal view of frameshift-induced *hoxb4a^{−/−};hoxb5a^{−/−};hoxb5b^{−/−}* and *hoxb4a-b5a^{del/del};hoxb5b^{del/del}* larvae. All images are captured from the dorsal side. Arrowheads indicate the presumptive positions of pectoral fin buds. (**M–R**) Expression patterns of zebrafish *hoxb4a*, *hoxb6a*, *hoxb6b*, *hoxb7a*, *hoxb8a*, and *hoxb8b* at the 10-somite stage. Dorsal views. Arrowheads indicate the presumptive positions of pectoral fin buds. Scale bars represent 100 μm.

The online version of this article includes the following figure supplement(s) for figure 7:

**Figure supplement 1.** Generation of locus-deletion mutants using CRISPR–Cas9.

a significant reduction in *tbx5a* expression when intercrossing *hoxba*$^{+/-}$;*hoxb5b*$^{+/del}$ fish (**Figure 7A, B, D, E**). However, the occurrence rate of embryos lacking pectoral fins ($n$ = 3/120; 2.5 %) was lower than predicted based on Mendelian genetics (1/16; 6.3 %). Other *hoxba*$^{-/-}$;*hoxb5b*$^{del/del}$ embryos did exhibit shortened pectoral fins (**Figure 7C**). Additionally, concerning the hoxba cluster, we also generated frameshift-induced *hoxb5a*$^{-/-}$;*hoxb5b*$^{-/-}$ and a full locus-deleted *hoxb5a*$^{del/del}$;*hoxb5b*$^{del/del}$ embryos (**Figure 7—figure supplement 1**). *hoxb5a*$^{-/-}$;*hoxb5b*$^{-/-}$ larvae consistently displayed normal pectoral fins (**Figure 7F**). In contrast, *hoxb5a*$^{del/del}$;*hoxb5b*$^{del/del}$ mutants exhibited a slightly more pronounced phenotype of shortened pectoral fins, but no instances of missing pectoral fins were identified (**Figure 7G**). To further clarify which Hox genes could potentially contribute to fin bud formation, we analyzed the expression of PG4 and PG6–8 genes from hoxba and hoxbb clusters using whole-mount in situ hybridization. Among these, only *hoxb4a* showed detectable expression in the fin buds, whereas *hoxb6a*, *hoxb6b*, *hoxb7a*, *hoxb8a*, and *hoxb8b* were not detectable (**Figure 7M–R**). Furthermore, due to the functional redundancy observed between *hoxb4a* and *hoxb5a* observed in zebrafish vertebral patterning and the RA-dependent expression of *hoxb4a* (**Grandel et al., 2002**; **Maeno et al., 2024**), we generated frameshift-induced *hoxb4a*$^{-/-}$;*hoxb5a*$^{-/-}$;*hoxb5b*$^{-/-}$ embryos and an allele lacking the entire genomic region encompassing *hoxb4a* and *hoxb5a* (**Figure 7—figure supplement 1**). Although *hoxb4a*$^{-/-}$;*hoxb5a*$^{-/-}$;*hoxb5b*$^{-/-}$ embryos did not lack pectoral fins (**Figure 7H**), some *hoxb4a-b5a*$^{del/del}$;*hoxb5b*$^{del/del}$ embryos did lack pectoral fins, although at low penetrance (**Figure 7I**, $n$ = 3/397; 0.7%). The majority of *hoxb4a-b5a*$^{del/del}$;*hoxb5b*$^{del/del}$ embryos displayed truncated pectoral fins (**Figure 7J**). The lack of pectoral fins was not detected in *hoxb4a-b5a*$^{del/del}$ and *hoxb4a*$^{del/del}$;*hoxb5b*$^{del/del}$ mutants (**Figure 7K, L**, **Figure 7—figure supplement 1**). These results suggest that *hoxb4a*, *hoxb5a*, and *hoxb5b* may cooperatively contribute to the anterior–posterior positioning of pectoral fin buds in zebrafish. Furthermore, our findings reveal that transcriptional adaptation may partially play a role in genetic compensation for frameshift-induced hox mutations during pectoral fin formation, while also suggesting the potential involvement of other unknown mechanisms or additional genomic regions in the initial induction of zebrafish pectoral fin buds.

## Discussion

Since the establishment of gene targeting (**Mansour et al., 1988**), numerous Hox knockout mice have been generated. However, neither single nor compound Hox mutants, nor Hox overexpression approaches, have been reported to cause significant defects in limb positioning (**Jurberg et al., 2013**; **Moreau et al., 2019**). Consequently, the role of Hox genes in the positioning of vertebrate paired appendages remains unclear due to a lack of direct genetic evidence. In this study, we provide genetic evidence using zebrafish, demonstrating that specific combinatorial deletions of hoxba and hoxbb clusters result in the absence of pectoral fins and a lack of induction of *tbx5a* expression in the pectoral fin field. Based on our results, we propose a model in which Hox expressions in the lateral plate mesoderm, regulated by RA synthesized in the paraxial mesoderm, provide positional information and induce *tbx5a* expression in the presumptive fin buds. Our genetic results shed light on longstanding fundamental questions regarding the molecular mechanisms by which the positioning of paired appendages is established in vertebrates.

### *hoxba/bb* clusters (possibly *hoxb4a*, *b5a*, *b5b*) define the pectoral fin field

We showed that *hoxba;hoxba* mutants exhibit a lack of pectoral fins. Abnormal expression of *tbx5a* in the lateral plate mesoderm, where pectoral fin precursor cells normally arise, was evident from its initial expression in the mutants. We presume that hox genes in hoxba and hoxbb clusters, induced by RA, define the region along the anterior–posterior axis where pectoral fin formation can occur and promote the expression of *tbx5a* in the lateral plate mesoderm. Importantly, *tbx5a* expression in the fin bud was not induced even by RA exposure in *hoxba;hoxbb* mutants (**Figure 4E–H**). In zebrafish, it has been suggested that RA synthesized in the paraxial mesoderm acts directly on the lateral plate mesoderm, or that RA induces *tbx5a* expression via *wnt2b* in the intermediate mesoderm (**Gibert et al., 2006**; **Grandel and Brand, 2011**; **Neto et al., 2012**). However, our results suggest that intact *hoxba* and *hoxbb* clusters are required for RA-dependent induction of *tbx5a* expression. One possible explanation is a linear pathway in which RA induces hox expression, which then activates

*tbx5a*. Alternatively, RA may induce hox expression and, together with Hox proteins, act cooperatively to activate *tbx5a*. Our data are consistent with both models. Through genetic studies, albeit with low penetrance, we further showed that *hoxb4a*, *hoxb5a*, and *hoxb5b* play a significant role in the induction of pectoral fin buds. The homologous Hoxb4 and Hoxb5 proteins in mice have been shown to bind directly to the enhancer of *Tbx5* (*Minguillon et al., 2012*). Moreover, in chickens, *Tbx5* expression in the forelimb field is restricted by posterior Hox genes, preventing posterior expansion (*Moreau et al., 2019*; *Nishimoto et al., 2014*). In contrast, our combinatorial deletions of zebrafish *hox* clusters did not produce posterior expansion of *tbx5a* expression. We therefore propose that, in zebrafish, *hoxb4a*, *hoxb5a*, and *hoxb5b*, induced by RA in the paraxial mesoderm, act primarily to promote *tbx5a* expression in the pectoral fin field.

## Partial phenocopy of locus-deleted hox mutants implies other unidentified mechanisms

In this study, we observed that hox mutants with deletions of genomic loci exhibited more pronounced pectoral fin phenotypes than frameshift-induced hox mutants. The frameshift mutants used in this study are likely to be loss-of-function alleles, as we demonstrated several defects in vertebral patterning in these mutants (*Maeno et al., 2024*). These results suggest that transcriptional adaptation may occur during the formation of pectoral fins in frameshift-induced hox mutants, potentially compensating for the loss of function. In our previous studies, which focused on dorsal and anal fin patterning, vertebral patterning, and swim bladder formation, we demonstrated that the phenotypes observed in zebrafish hox cluster-deleted mutants could essentially be replicated by introducing frameshift mutations into hox genes within the cluster (*Adachi et al., 2024*; *Maeno et al., 2024*; *Satoh et al., 2025*; *Yamada et al., 2021*). Therefore, creating a mutant with a deleted locus that does not induce transcriptional adaptation was unnecessary. One factor that may explain the differences in results between this study and our prior findings is that even a small amount of hox gene products essential for pectoral fin positioning may be sufficient to induce fin buds, as shown in *Figure 1*, potentially leading to the manifestation of gene compensation through transcriptional adaptation in frameshift-induced mutants. Alternatively, transcriptional adaptation may function differently depending on various developmental processes or stages. Interestingly, all the aforementioned phenotypes reproducible by frameshift mutations were observed at later developmental stages than pectoral fin formation. Additionally, in hox locus-deleted mutants, only some *hoxba*$^{-/-}$;*hoxb5b*$^{de/dell}$ or *hoxb4a-b5a*$^{del/del}$;*hoxb5b*$^{del/del}$ embryos were able to reproduce the loss of the pectoral fin, suggesting the presence of other compensatory mechanisms. The organized expression patterns of Hox genes are thought to be regulated by complex regulatory controls, with their expression regulated according to the genomic order within the cluster. In the absence of Hox function, neighboring intact Hox genes could compensate for the loss of function. It is also possible that unidentified non-coding regions in hoxba and hoxbb clusters are involved in pectoral fin positioning. These issues should be clarified in future studies.

## Hox genes and the evolutionary origin of pectoral fins

According to fossil records from the Cambrian period, the earliest vertebrates did not possess paired fins (*Janvier, 1996*; *Shubin et al., 1997*). Later in evolution, vertebrates acquired primitive pectoral fins, which served as evolutionary precursors to tetrapod forelimbs, before diverging into ray- and lobe-finned fishes. Our results align well with this evolutionary scenario. Hox genes, which are conserved across bilaterian animals, were already present in primitive vertebrates without paired fins, suggesting that their functional diversification may have facilitated the origin of new appendages. The early fossils of vertebrates with pectoral fins were discovered in Ordovician and Silurian jawless fishes (*Janvier, 1996*; *Shubin et al., 1997*). Our finding that *hoxba*;*hoxbb* mutants completely lack pectoral fins implies that molecular evolution within the ancestral HoxB cluster contributed to the emergence of pectoral fins. Although the precise molecular modifications remain unclear, one plausible model is that HoxB genes acquired the capacity to induce *Tbx5* expression in the lateral plate mesoderm, establishing the RA–HoxB–Tbx5 gene network essential for fin initiation. We presume that the acquisition of pectoral fins was driven by multiple genomic changes accumulated over time, among which molecular evolution within the HoxB cluster may have represented one of the principal factors contributing to the origin of this novel appendage.

## The positioning of pectoral fins is restricted to specific hox clusters in zebrafish, in contrast to the functional overlap seen in mice

Taking advantage of our availability of seven individual hox cluster mutants in zebrafish, we generated multiple cluster deletions and demonstrated that hox genes required for establishing pectoral fin buds are restricted to hoxba and hoxbb clusters. In contrast, analyses of knockout mice have revealed extensive functional overlap among the four Hox clusters. For instance, mice lacking all HoxB genes except for *Hoxb13* do not lose forelimbs (*Medina-Martínez et al., 2000*), and forelimbs are still present in *Hoxa5;Hoxb5;Hoxc5* triple knockout mice, which lack all Hox genes of the paralog group 5 (*Xu et al., 2013*). Thus, while zebrafish *hoxba;hoxbb* mutants exhibit a striking phenotype of absent pectoral fins, forelimbs are preserved in comparable mouse mutants. We speculate that paired fins emerged after the quadruplication of primitive Hox clusters, and that novel functions related to pectoral fin formation arose within the HoxB cluster. In zebrafish, these functions remain confined to hoxba and hoxbb clusters, making them indispensable for pectoral fin development. Interestingly, a similar cluster-specific confinement has been observed in median fins: in both zebrafish and medaka, Hox genes responsible for positioning dorsal and anal fins are primarily located in HoxC-related clusters (*Adachi et al., 2024*). In contrast, in mice, functional redundancy among Hox clusters—likely accumulated during over half a billion years of vertebrate evolution—may mask the requirement of HoxB genes for forelimb formation. While the precise evolutionary mechanisms remain to be clarified, our results suggest that cluster-specific specialization in teleosts contrasts with broader redundancy across Hox clusters in mammals.

# Materials and methods

## Zebrafish

Riken Wild-type (RW) zebrafish, obtained from the National BioResource Project in Japan, were maintained at 27°C with a 14-hr light/10-hr dark cycle. Embryos were collected from natural spawning, and the larvae were raised at 28.5°C. Developmental stages of the embryos and larvae were determined based on hours post-fertilization (hpf), days post-fertilization (dpf), and developmental stage-specific features (*Kimmel et al., 1995*; *Parichy et al., 2009*). The alleles of *hox* cluster-deleted mutants used in this study were previously generated using CRISPR–Cas9 (*Yamada et al., 2021*). The following frameshift-induced *hox* mutants were previously created using CRISPR–Cas9: *hoxb5a*$^{sud125}$, *hoxb5b*$^{sud136}$, *hoxb6b*$^{sud137}$, and *hoxb4a;hoxb5a*$^{sud144}$ (*Maeno et al., 2024*). All experiments involving live zebrafish were approved by the Committee for Animal Care and Use of Saitama University.

## Generation of mutants by the CRISPR–Cas9 system

All the mutants used in this study were generated using the Alt-R CRISPR–Cas9 system (Integrated DNA Technologies). To create frameshift-induced mutants, gene-specific crRNAs were incubated with common tracrRNA, followed by Cas9 nuclease. For the generation of mutants that delete large genomic regions, two crRNAs targeting both ends of the regions were incubated with common tracrRNA and Cas9 nuclease. Approximately one nanoliter of the crRNA:tracrRNA-Cas9 RNA–protein complex was injected into fertilized embryos, which were subsequently raised to juvenile fish. Candidate founder fish were selected using a heteroduplex mobility shift assay for frameshift mutants (*Ota et al., 2013*) and by amplifying genomic deletions with specific primers. After sexual maturation, candidate founder fish were mated with wild-type fish to produce heterozygous F1 offspring. Among the F1 offspring, mutant fish carrying the same mutation were identified through PCR-based genotyping followed by DNA sequencing. The target-specific sequences of the crRNAs used in this study are listed (*Supplementary file 1B*). The frozen sperm from the mutants used in this study have been deposited in the National BioResource Project Zebrafish in Japan (https://shigen.nig.ac.jp/zebra/) and are available upon request.

## Genotyping of mutants

For the phenotypic analysis of embryos and the maintenance of mutant fish lines, PCR-based genotyping was performed. Briefly, genomic DNA was extracted from the analyzed embryos or the partially dissected caudal fins of anesthetized larvae or fish using the NaOH method (*Meeker et al., 2007*), which served as the template for PCR. The sequences of the primers used for genotyping

frameshift-induced mutants are listed in the *Supplementary file 1C*. For genotyping deletion mutants of *hox* genes, PCR was conducted using a combination of three primers, with their sequences also provided (*Supplementary file 1C*). Genotyping of *hox* cluster-deleted mutants was performed using PCR as previously described (*Yamada et al., 2021*). After the reactions, the PCR products were separated by electrophoresis. Based on differences in the lengths of PCR products derived from wild-type and mutated alleles, either a 2% agarose gel in 0.5x TBE buffer, a 15% polyacrylamide gel in 0.5x TBE buffer, or direct DNA sequencing was utilized to determine the genotype.

## Whole-mount in situ hybridization

Whole-mount in situ hybridization was carried out as described (*Thisse and Thisse, 2014*). After the staining, the embryos mounted in 70–80% glycerol were captured under the stereomicroscope (Leica M205 FA) with a digital camera (Leica DFC350F). After taking images, PCR-based genotyping was carried out as described above.

## RNA-seq analysis

For RNA-seq analysis, embryos were obtained by the intercrosses between *raldh2* heterozygous fish. At the 18- to 20-somite stages, sibling and homozygous embryos were separated based on the phenotype, and 20 embryos per tube were collected. Then, total RNA was extracted by ISOGENE (Nippon Gene) and followed by DNase I treatment (Takara). After the quality check of the isolated RNA, RNA-seq analyses ($n = 2$ for each) were carried out by Genewiz, Azenta Life Science.

## Treatment of RA in the embryos

Treatment of RA on the embryos was performed as previously described, with minor modifications (*Gibert et al., 2006*; *Grandel et al., 2002*). Briefly, embryos were obtained from intercrosses between *raldh2* heterozygous mutants. At the 70% epiboly stage, the embryos were soaked in a solution containing DMSO or $10^{-8}$ M RA and incubated at 28.5°C. Embryos were fixed at 30 hpf for analysis of *tbx5a* expression and at the 10-somite stage for examination of *hox* gene expression.

## Acknowledgements

We thank the NBRP zebrafish for providing fish and preserving the mutant lines generated in this study. We also appreciate Drs. Koji Tamura and Gembu Abe for providing the zebrafish *tbx5a* plasmid for in situ hybridization. This work was supported by KAKENHI Grants-in-Aid for Scientific Research from the Ministry of Education, Culture, Sports, Science, and Technology, Japan (18K06177, 23K05790 to AK).

## Additional information

### Funding

| Funder | Grant reference number | Author |
| --- | --- | --- |
| KAKENHI | 18K06177 | Akinori Kawamura |
| KAKENHI | 23K05790 | Akinori Kawamura |

The funders had no role in study design, data collection, and interpretation, or the decision to submit the work for publication.

### Author contributions

Morimichi Kikuchi, Conceptualization, Formal analysis, Investigation; Renka Fujii, Daiki Kobayashi, Yuki Kawabe, Haruna Kanno, Sohju Toyama, Farah Tawakkal, Kazuya Yamada, Investigation; Akinori Kawamura, Conceptualization, Supervision, Investigation, Writing – original draft, Project administration, Writing – review and editing

### Author ORCIDs

Yuki Kawabe ⓘ https://orcid.org/0009-0008-9149-1605
Haruna Kanno ⓘ https://orcid.org/0009-0008-5751-5992

Sohju Toyama https://orcid.org/0009-0006-8684-8345
Farah Tawakkal https://orcid.org/0009-0004-4438-7638
Akinori Kawamura https://orcid.org/0000-0002-5618-7113

### Ethics

All experiments involving live zebrafish were approved by the Committee for Animal Care and Use of Saitama University (R2-A-1-6, R3-A-1-6, R4-A-1-7, R5-A-1-7, R6-A-1-7) and were conducted in accordance with the Animal Research Reporting of In Vivo Experiments (ARRIVE) guidelines as well as all relevant institutional and national regulations.

Reviewer #1 (Public review): https://doi.org/10.7554/eLife.105889.3.sa1
Reviewer #2 (Public review): https://doi.org/10.7554/eLife.105889.3.sa2
Author response https://doi.org/10.7554/eLife.105889.3.sa3

---

# Additional files

### Supplementary files

Supplementary file 1. The frameshift mutations introduced in *hoxbb* genes in the embryos lacking pectoral fins during the screening. The DNA sequences of *hoxb5b*, *hoxb6b*, and *hoxb8b* were analyzed in 2 dpf embryos apparently lacking pectoral fins. Among the CRISPR–Cas9-induced mutations, those resulting in frameshifts are highlighted in blue.

Supplementary file 2. The target-specific sequences of crRNAs used in this study.

Supplementary file 3. The sequences of primers used for the genotyping in this study.

MDAR checklist

### Data availability

All data necessary to support the conclusions of this article are provided within the main text, figures, and supplementary files.

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
