## [Editor Report · eLife Assessment]

This **important** study advances our understanding of vertebrate forelimb development, specifically the contribution of Hox genes to zebrafish pectoral fin formation. The authors have employed a robust and extensive genetic approach to tackle a key and unresolved question. The findings are overall **convincing** and will be of broad interest to developmental and evolutionary biologists.

---

## [Referee Report · Reviewer #1 (Public review)]

Summary:

The authors have used gene deletion approaches in zebrafish to investigate the function of genes of the hox clusters in pectoral fin "positioning" (but perhaps more accurately pectoral fin "formation")

Strengths:

The authors have employed a robust and extensive genetic approach to tackle an important and unresolved question.

The results are largely very clearly presented.

Weaknesses:

The Abstract suggests that no genetic evidence exists in model organisms for a role of Hox genes in limb positioning. There are, however, several examples in mouse and other models (both classical genetic and other) providing evidence for a role of Hox genes in limb position, which is elaborated on in the Introduction.

It would perhaps be more accurate to state that several lines of evidence in a range of model organisms (including the mouse) support a role for Hox genes in limb positioning. The author's work is not weakened by a more inclusive introduction that cites the current literature more comprehensively.

It would be helpful for the authors to make a clear distinction between "positioning" of the limb/fin and whether a limb/fin "forms" at all, independent of the relative position of this event along the body axis.

Discussion of why the zebrafish is sensitive to Hoxb loss with reference to the fin, but mouse Hoxb mutants do make a limb?

Is this down to exclusive expression of Hoxbs in the zebrafish pectoral fin forming region rather than a specific functional role of the protein? This is important as it has implications for the interpretation of results throughout the paper and could explain some apparently conflicting results. .

Why is hoxba more potent than hoxbb? Is this because Hioxba has Hox4/5 present while hox bb has only hoxb5? Hoxba locus has retained many more hox genes in,cluater than hoxbb therefore might expect to see greater redundancy in this locus

Deletion of either hox a or hox d in background of hoxba mutant does have some effect. IS this a reflection of protein function or expression dynamics of hoax/hoxd genes?

Can we really be confident there is a "transformation of pectoral fin progenitor cells into cardiac cells"?

The failure to repress Nkx2.5 in the posterior (pelvic fin) domain is clear but have these cells actually acquired cardiac identity? They would be expected to express Tbx5a (or b) as cardiac precursors but this domain does not broaden. There is no apparent expansion of the heart (field)/domain or progenitors beyond 16 somite stage. The claimed "migration" of heart precursors iin the mutant is not clear. The heart/cardiac domain that does form in the mutant is not clearly expanded in the mutant. The domain of cmlc2 looks abnormal in the mutant but I am not convinced it is "enlarged" as claim by the authors. The authors have not convincingly shown that " the cells that should form the pectoral fin instead differentiate into cardia cells."

The only clear conclusion is the loss of pectoral fin-forming cells rather than these fin-forming cells being "transformed" into a new identity. It would be interesting to know what has happened to the cells of the pectoral fin forming region in these double mutants.

It is not clear what the authors mean by a "converse" relationship between forelimb/pectoral fin and heart formation. The embryological relationship between these two populations is distinct in amniotes.

The authors show convincing data that RA cannot induce Tbx5a in the absence of Hob clusters but I am not convinced by the interpretation of this result. The results shown would still be consistent with RA acting directly upstream of tbx5a but merely that RA acts in concert with hox genes to activate tbx5a. IN the absence of one or the other tbx5a would not be expressed. It is not necessary that RA and hoxbs act exclusively in a linear manner (i.e. RA regulates hoxb that in turn regulate tbx5a)

The authors have carried out a functional test for the function of hoxb6 and hoxb8 in the hemizygous hoxb mutant background. What is lacking is any expression analysis to demonstrate whether hoxb6b or hoxb8b are even expressed in the appropriate pectoral fin territory to be able to contribute to pectoral fin development either in this assay or in normal pectoral fin development.

(The term "compensate" used in this section is confusing/misleading.)

The authors' confounding results described in Figures 6-7 are consistent with the challenges faced in other model organisms in trying to explore the function of genes in the hox cluster and the known redundancy that exists across paralogous groups and across individual clusters.

Given the experimental challenges in deciphering the actual functions of individual or groups of hox genes, a discussion of the normal expression pattern of individual and groups of hox genes (and how this may change in different mutant backgrounds) could be helpful to make conclusions about likely normal function of these genes and compensation/redundancy in different mutant scenarios.

Comments on revisions:

No further issues to address.

---

## [Referee Report · Reviewer #2 (Public review)]

Summary:

The authors of this manuscript performed a fascinating set of zebrafish mutant analysis on hox cluster deletion and pinpoint the cause of the pectoral fin loss in one combinatorial hox cluster mutant of hoxba and hoxbb. I support the publication of this manuscript.

Strengths:

The study is based on a variety of existing experimental tools that enabled the authors' past construction of hox cluster mutants and is well-designed. The manuscript is well written to report the author's findings on the mechanism that positions the pectoral fin.

Weaknesses:

The study does not focus on the other hox clusters than ba and bb, and is confined to the use of zebrafish, as well as the comparison with existing reports from mouse experiments.

Comments on revisions:

The authors have sufficiently addressed the concerns raised in my previous review. The revised manuscript substantially strengthens the original work.

---

## [Author Response]

The following is the authors’ response to the original reviews.

**Reviewer #1 (Public review):**
Summary:The authors have used gene deletion approaches in zebrafish to investigate the function of genes of the hox clusters in pectoral fin "positioning" (but perhaps more accurately pectoral fin "formation").Strengths:The authors have employed a robust and extensive genetic approach to tackle an important and unresolved question. The results are largely presented in a very clear way.We thank the reviewer for the positive summary and for recognizing the strengths of our genetic approach and presentation.Weaknesses:The Abstract suggests that no genetic evidence exists in model organisms for a role of Hox genes in limb positioning. There are, however, several examples in mouse and other models (both classical genetic and other) providing evidence for a role of Hox genes in limb position, which is elaborated on in the Introduction.It would perhaps be more accurate to state that several lines of evidence in a range of model organisms (including the mouse) support a role for Hox genes in limb positioning. The author's work is not weakened by a more inclusive introduction that cites the current literature more comprehensively.

Thank you for this constructive comment. We agree that our Abstract implied an absence of genetic evidence across model organisms and could be misleading. We have revised the Abstract to acknowledge that multiple lines of evidence—including classical and molecular studies in mouse and other models—support a role for Hox genes in limb/fin positioning. We have also expanded the Introduction to cite this literature more comprehensively. These changes clarify the current state of knowledge while preserving the novelty of our zebrafish genetic findings.

It would be helpful for the authors to make a clear distinction between "positioning" of the limb/fin and whether a limb/fin "forms" at all, independent of the relative position of this event along the body axis.

We thank the reviewer for pointing this out. In the revised manuscript, we now make a distinction between these two aspects: we describe “positioning” as being specified by the expression domains of Hox genes along the anterior–posterior axis, while the “formation” of pectoral fins reflects the functional requirement of Hox genes to induce tbx5a expression and thereby initiate fin development. We have clarified this distinction in the text to better separate these related but distinct roles of Hox genes.

Discussion of why the zebrafish is sensitive to Hoxb loss with reference to the fin, but mouse Hoxb mutants do make a limb?

We thank the reviewer for this important comment. Our interpretation is that paired fins first appeared in vertebrates that already possessed four Hox clusters. It is likely that novel functions related to pectoral fin positioning emerged within the HoxB cluster at that time, contributing to the origin of pectral fins. In zebrafish, we found that these functions remain largely restricted to the hoxba and hoxbb clusters, such that loss of both results in complete absence of pectoral fins. In contrast, mice exhibit a high degree of functional redundancy across Hox clusters. For example, deletion of all HoxB genes except Hoxb13 does not result in forelimb loss (Medina-Martinez et al., 2000), and forelimbs are still present in Hoxa5;Hoxb5;Hoxc5 triple knockouts (Xu et al., 2013). Thus, although we cannot fully explain why HoxB cluster deletions alone do not abolish forelimb formation in mice, it is plausible that overlapping functions from other Hox clusters compensate for the loss of HoxB genes, consistent with the general robustness of the mammalian Hox system. We have revised the Discussion to clarify this point.

Is this down to exclusive expression of Hoxbs in the zebrafish pectoral fin forming region rather than a specific functional role of the protein? This is important as it has implications for the interpretation of results throughout the paper and could explain some apparently conflicting results.

We thank the reviewer for this insightful comment. To address this point, we newly analyzed the expression patterns of PG4–8 genes in the hoxba and hoxbb clusters. Our in situ hybridization results revealed that only hoxb4a, hoxb5a, and hoxb5b are detectably expressed in the pectoral fin buds (Figure 5C, 5E, Figure 7M-R). While we cannot completely exclude the possibility of functional differences among Hox proteins, our data strongly suggest that the loss of pectoral fins in hoxba;hoxbb cluster mutants is primarily due to the expression domains of these specific Hox genes in the fin-forming region, rather than to unique biochemical properties of the proteins. We have added these new data as a figure in the revised manuscript (Figure 7M-R) and clarified this point in the text (lines 312-316).

Why is Hoxba more potent than Hoxbb? Is this because Hoxba has Hox4/5 present, while Hoxbb has only Hoxb5? Hoxba locus has retained many more Hox genes in cluster than hoxbb; therefore, one might expect to see greater redundancy in this locus.

We thank the reviewer for raising this important point. At present, we do not know the precise reason why hoxba appears more potent than hoxbb. The possibility raised by the reviewer—that differences in retained gene content (e.g., Hox4/5 in hoxba versus only Hoxb5 in hoxbb) may underlie this discrepancy—is certainly plausible. However, our previous study on the formation of dorsal and anal fins showed a similar situation: although PG11–13 Hox genes are present in both hoxca and hoxcb clusters, deletion of hox genes in hoxca cluster had a more pronounced effect on median fin development (Adachi et al., 2024). This suggests that, following the teleost-specific whole-genome duplication, duplicated Hox clusters are not functionally equivalent, and asymmetric retention or deployment of functions may occur. The mechanistic basis of such bias remains unclear and warrants further investigation.

Deletion of either Hoxa or Hoxd in the background of the Hoxba mutant does have some effect. Is this a reflection of protein function or expression dynamics of Hoxa/Hoxd genes?

We appreciate the reviewer’s comment and the opportunity to clarify this point. In Figure 2, we compared several double mutants with the hoxba single mutant. Among thesm, only the hoxba;hoxbb mutant exhibited a complete loss of tbx5a expression, whereas other combinations did not differ substantially from the hoxba mutant alone. Therefore, we consider that additional deletions such as hoxaa, hoxab, and hoxda do not have a strong effect beyond the hoxba deletion itself, and it is unlikely that Hoxa or Hoxd proteins functionally compensate for Hoxba in regulating tbx5a expression. Consistent with this interpretation, in our previous study we did not detect abnormalities in tbx5a expression in the hoxaa;hoxab;hoxda triple mutant (Ishizaka et al., 2024). Taken together, these observations support our view that the hoxba and hoxbb clusters are specifically required for the induction of tbx5a in the pectoral fin field.

Can we really be confident that there is a "transformation of pectoral fin progenitor cells into cardiac cells"?The failure to repress Nkx2.5 in the posterior (pelvic fin) domain is clear, but have these cells actually acquired cardiac identity? They would be expected to express Tbx5a (or b) as cardiac precursors, but this domain does not broaden. There is no apparent expansion of the heart (field)/domain or progenitors beyond the 16 somite stage. The claimed "migration" of heart precursors in the mutant is not clear. The heart/cardiac domain that does form in the mutant is not clearly expanded in the mutant. The domain of cmlc2 looks abnormal in the mutant, but I am not convinced it is "enlarged" as claimed by the authors. The authors have not convincingly shown that "the cells that should form the pectoral fin instead differentiate into cardiac cells." The only clear conclusion is the loss of pectoral fin-forming cells rather than these fin-forming cells being "transformed" into a new identity. It would be interesting to know what has happened to the cells of the pectoral fin-forming region in these double mutants.

We sincerely thank the reviewer for this important comment. We agree that our data do not yet allow us to conclude with certainty that the presumptive pectoral fin progenitor cells in hoxba;hoxbb cluster mutants are fully “transformed” into cardiac cells. Our intention was to describe the striking posterior expansion of nkx2.5 expression and the altered morphology of the cmlc2-positive cardiac field in the mutants, which suggested a shift in cell fate. However, as the reviewer correctly points out, we did not directly demonstrate that the missing fin progenitors acquire bona fide cardiac identity.

To address this, we have revised the text to clarify that the most robust conclusion from our current dataset is the loss of pectoral fin-forming cells in hoxba;hoxbb cluster mutants. We have softened or removed the claim of “transformation” and instead emphasize that our observations are consistent with an expansion of cardiac marker expression domains into the region where fin progenitors normally arise. We also acknowledge that the cmlc2 domain is abnormal rather than unequivocally enlarged, and have adjusted our wording accordingly.

It is not clear what the authors mean by a "converse" relationship between forelimb/pectoral fin and heart formation. The embryological relationship between these two populations is distinct in amniotes.

We thank the reviewer for pointing this out. Our intention was to highlight the reciprocal balance between pectoral fin and cardiac progenitors in zebrafish. In particular, Waxman et al. (2008) demonstrated that retinoic acid signaling promotes pectoral fin formation while restricting the expansion of cardiac progenitors, thereby illustrating this reciprocal relationship. To avoid confusion, we have revised the text to explicitly state that this applies to zebrafish.

The authors show convincing data that RA cannot induce Tbx5a in the absence of Hob clusters, but I am not convinced by the interpretation of this result. The results shown would still be consistent with RA acting directly upstream of tbx5a, but merely that RA acts in concert with hox genes to activate tbx5a. In the absence of one or the other, Tbx5a would not be expressed. It is not necessary that RA and hoxbs act exclusively in a linear manner (i.e., RA regulates hoxb that in turn regulates tbx5a).

We appreciate the reviewer’s thoughtful comment. We agree that our original wording in the Results section implied a strictly linear model of RA→Hox→tbx5a. In response, we have revised the Results to state only the experimental observation, namely that RA-dependent induction of tbx5a does not occur in the absence of the hoxba and hoxbb clusters.

We have moved the broader interpretation to the Discussion, where we now emphasize that our data are compatible with multiple models. One possibility is a linear pathway in which RA induces Hox expression that subsequently activates tbx5a. Alternatively, it is also plausible that RA induces Hox expression and that RA and Hox proteins act cooperatively to induce tbx5a. Our findings do not distinguish between these possibilities, and both models remain consistent with the data. We believe this restructuring addresses the reviewer’s concern by keeping the Results factual and limiting mechanistic interpretation to the Discussion.

The authors have carried out a functional test for the function of hoxb6 and hoxb8 in the hemizygous hoxb mutant background. What is lacking is any expression analysis to demonstrate whether Hoxb6b or Hoxb8b are even expressed in the appropriate pectoral fin territory to be able to contribute to pectoral fin development, either in this assay or in normal pectoral fin development.

We thank the reviewer for emphasizing the importance of expression analyses. In response, we performed a comprehensive whole-mount in situ hybridization survey of all eight PG4–8 Hox genes from the hoxba and hoxbb clusters (hoxb4a, hoxb5a, hoxb5b, hoxb6a, hoxb6b, hoxb7a, hoxb8a, and hoxb8b) during pectoral fin development (18–30 hpf). Among these, only hoxb4a, hoxb5a, and hoxb5b displayed detectable expression in the developing pectoral fin buds. In contrast, hoxb6a, hoxb6b, hoxb7a, hoxb8a, and hoxb8b were not expressed in this territory. These new data have been incorporated into the revised manuscript (Fig. 7M-R). We believe that this dataset provides a more complete and systematic picture of which Hoxb genes are available to function in pectoral fin development, and we are grateful to the reviewer for this valuable suggestion, which significantly strengthened our study.

(The term "compensate" used in this section is confusing/misleading.)

We thank the reviewer for this helpful remark. We agree that the term “compensate” was misleading in this context, as it could be confused with genetic compensation mechanisms such as transcriptional adaptation. To avoid this ambiguity, we have revised the wording.

Specifically, we replaced “compensate for” with “mimic the effect of” or “phenocopy” depending on the context. We believe this revision improves clarity and prevents misunderstanding.

The authors' confounding results described in Figures 6-7 are consistent with the challenges faced in other model organisms in trying to explore the function of genes in the hox cluster and the known redundancy that exists across paralogous groups and across individual clusters. Given the experimental challenges in deciphering the actual functions of individual or groups of hox genes, a discussion of the normal expression pattern of individual and groups of hox genes (and how this may change in different mutant backgrounds) could be helpful to make conclusions about likely normal function of these genes and compensation/redundancy in different mutant scenarios.

We appreciate the reviewer’s thoughtful comment. We agree that functional analyses of Hox genes are often complicated by redundancy within and across clusters. In this revision, we have included additional expression data of PG4–8 genes from the hoxba and hoxbb clusters, showing that only hoxb4a, hoxb5a, and hoxb5b are expressed in the fin buds. Although we did not analyze expression changes across mutant backgrounds in this study, we consider this an important direction for future experiments.

**Reviewer #2 (Public review):**
Summary:The authors of this manuscript performed a fascinating set of zebrafish mutant analyses on hox cluster deletion and pinpointed the cause of the pectoral fin loss in one combinatorial hox cluster mutant of Hoxba and Hoxbb.Strengths:The study is based on a variety of existing experimental tools that enabled the authors' past construction of hox cluster mutants, and is well-designed. The manuscript is well written to report the authors' findings on the mechanism that positions the pectoral fin.Weaknesses:The study does not focus on the other hox clusters other than ba and bb, and is confined to the use of zebrafish, as well as the comparison with existing reports from mouse experiments.

We thank the reviewer for the thoughtful and encouraging evaluation of our manuscript. We are pleased that the strengths of our study design and clarity of writing were recognized. We also acknowledge the noted limitations, and while our focus here is on zebrafish hoxba and hoxbb clusters, we agree that future studies should expand to other hox clusters and additional models. Below, we provide individual responses to the specific points raised.

**Reviewer #1 (Recommendations for the authors):**
(1) Some additional expression analyses of Hoxb6/b8 etc, could be carried out to address some issues raised in the main review.

We thank the reviewer for this suggestion. In response, we performed additional whole-mount in situ hybridization analyses of PG4–8 genes from the hoxba and hoxbb clusters, including hoxb6b and hoxb8b. These experiments showed that only hoxb4a, hoxb5a, and hoxb5b are expressed in the developing fin buds, whereas hoxb6a, hoxb6b, hoxb7a, hoxb8a, and hoxb8b are not. We have incorporated these new data into the revised manuscript (Figure 7M-R), which we believe clarify why functional tests of hoxb6b and hoxb8b did not uncover specific requirements in fin development.

(2) The discussion section, particularly the more speculative section on evolutionary significance, could be reduced. Discussion of pelvic fin could be removed also, as this has not and could not be addressed with the current experimental design.

We thank the reviewer for this helpful suggestion. In line with the recommendation, we have reduced the speculative section on evolutionary significance in the Discussion to make it more concise and focused. We have also removed the discussion of pelvic fins, as these were not directly addressed by our current experimental design. We believe these changes improve the clarity and focus of the Discussion section.

(3) The conclusions on transformation to cardiac identity could be reevaluated and presented differently.

We appreciate the reviewer’s insightful comment. In the revised manuscript, we have toned down our interpretation regarding a transformation to cardiac identity. Instead, we now describe the findings more cautiously, emphasizing the clear loss of fin precursors rather than a definitive acquisition of cardiac fate. We believe this revision presents a more balanced interpretation of the data.

(4) Minor typographical - I would suggest removing "Genetic Evidence:" from the title.

We appreciate the reviewer’s suggestion. In accordance with this comment, we have revised the title to: “HoxB-derived hoxba and hoxbb clusters are essential for the anterior-posterior positioning of zebrafish pectoral fins”.

**Reviewer #2 (Recommendations for the authors):**
(1) The authors mention the redundancy (between the a type and b type) of Hox clusters derived from an additional whole genome duplication in the teleost fish lineage. But, they do not refer to whether the zebrafish Tbx5 ortholog has an additional copy. This information helps the readers' interpretation of the data presented. First of all, tbx5a suddenly appears on line 143 without introducing its relationship with Tbx5, which needs to be explained in a revised manuscript.

We thank the reviewer for highlighting this important point. In zebrafish, there are indeed two Tbx5 orthologs, tbx5a and tbx5b. In the revised manuscript, we have modified the text around line 124 to introduce tbx5a in the context of its orthology to Tbx5, ensuring that its appearance in the Results is clear to the readers.

(2) I did not readily get whether the limb/fin 'positioning' that the authors focus on in this study is 'anteroposterior' positioning, but not anything else. If it is what is meant, the word 'anteroposterior' should just be inserted at the first appearance of the word 'positioning'.

We thank the reviewer for pointing this out. Our study specifically addresses the anteroposterior positioning of paired appendages, that is, how the initial site of pectoral fin formation is defined along the anterior–posterior axis of the body. To clarify this, we have revised the text to insert the word “anteroposterior” at the first appearance of the term “positioning” in both the Abstract and Introduction (lines 26 and 53). We believe this change resolves the ambiguity and makes the focus of our study explicit.

(3) Figure 5B also shows the remarkable reduction of hoxc1a expression, which the authors do not mention at all. I wonder how this is explained and how the authors justify no remark on this throughout the manuscript.

We thank the reviewer for this insightful comment. As correctly noted, we did observe a marked reduction of hoxc1a expression in Figure 5B. However, based on our genetic analyses, we consider that the causal genes underlying the phenotype are most likely located in hoxba and hoxbb clusters. Therefore, although the change in hoxc1a expression is indeed a notable phenomenon, we did not emphasize it in the manuscript in order to maintain focus on the primary clusters responsible for the observed phenotype (lines 240-241). We agree that this point should be acknowledged, and we have now added a brief note in the Results to clarify our findings.

(4) Figure 1 consists of multiple panels (A-M) but lacks panel D.

We apologize for the oversight. We have corrected it.

(5) Line 85 - precise role -> exact role.

We have corrected it (line 95).

(6) Line 87 - the vertebrate class Actinopterygii & the class Sarcopterygii.

Thank the reviewer for pointing out. We have corrected it (line 98-99).

(7) Line 90 - homologous -> orthologous.

We have corrected it (line 102).

(8) Figure 5 - For interpretability of the data, I suggest writing 'Paralogous groups' on the top of the panels A and B, and 'Cluster' vertically on the left.

We thank the reviewer for this helpful suggestion. As recommended, we have added

“Paralogous groups” at the top of panels A and B, and “Clusters” vertically on the left side of Figure 5 to facilitate interpretation of the data.

(9) Some subheading titles are too long. They can be shortened into 'hoxb5a and -b5b expression in pectoral fin buds are RA-dependent' instead of 'Expression patterns of hoxb5a and hoxb5b in pectoral fin buds are dependent on RA', for example.

We appreciate the reviewer’s suggestion regarding the length of the subheading titles. In response, we have shortened the relevant subheadings in both the Results and Discussion sections to make them more concise while retaining their scientific meaning. For example, the subheading originally written as “Expression patterns of hoxb5a and hoxb5b in pectoral fin buds are dependent on RA” has been revised to “hoxb5a/b5b expression in pectoral fin buds is

RA-dependent.” Similar adjustments have been made to other subheadings throughout these sections. We believe these changes improve readability and consistency without altering the intended content.

(10) Line 408 - why tetrapods, instead of cartilaginous fishes, which are thought of as natural in this context?

We appreciate the reviewer’s careful reading and insightful comment. However, in response to Reviewer 1’s suggestion, we have substantially reduced the speculative section on evolutionary significance in the Discussion. As a result, this specific part of the text has now been deleted. We thank the reviewer for raising this point.